# A Systematic and Critical Review on the Research Landscape of Finance in Vietnam from 2008 to 2020

**Manh-Tung Ho** [1], **Ngoc-Thang B. Le** [2], **Hung-Long D. Tran** [2], **Quoc-Hung Nguyen** [3], **Manh-Ha Pham** [4], **Minh-Hoang Ly** [4], **Manh-Toan Ho** [1], **Minh-Hoang Nguyen** [1] **and Quan-Hoang Vuong** [1,*]

[1] Centre for Interdisciplinary Social Research, Phenikaa University, Yen Nghia Ward, Ha Dong District, Hanoi 100803, Vietnam; tung.homanh@phenikaa-uni.edu.vn (M.-T.H.); toan.homanh@phenikaa-uni.edu.vn (M.-T.H.); hoang.nguyenminh@phenikaa-uni.edu.vn (M.-H.N.)
[2] College of Asia Pacific Studies, Ritsumeikan Asia Pacific University, Beppu, Oita 874-8577, Japan; buinle18@apu.ac.jp (N.-T.B.L.); tr19d6dn@apu.ac.jp (H.-L.D.T.)
[3] College of International Management, Ritsumeikan Asia Pacific University, Beppu, Oita 874-8577, Japan; ng19q6lh@apu.ac.jp
[4] BeeKrowd Corporation, Deutsches Haus, 33 Le Duan, District 1, Ho Chi Minh City 710000, Vietnam; ha.pham@beekrowd.com (M.-H.P.); hoang.ly@beekrowd.com (M.-H.L.)
[*] Correspondence: hoang.vuongquan@phenikaa-uni.edu.vn

**Abstract:** This paper endeavors to understand the research landscape of finance research in Vietnam during the period 2008 to 2020 and predict the key defining future research directions. Using the comprehensive database of Vietnam's international publications in social sciences and humanities, we extract a dataset of 314 papers on finance topics in Vietnam from 2008 to 2020. Then, we apply a systematic approach to analyze four important themes: Structural issues, Banking system, Firm issues, and Financial psychology and behavior. Overall, there have been three noticeable trends within finance research in Vietnam: (1) assessment of financial policies or financial regulation, (2) deciphering the correlates of firms' financial performances, and (3) opportunities and challenges in adopting innovations and ideas from foreign financial market systems. Our analysis identifies several fertile areas for future research, including financial market analysis in the post-COVID-19 eras, fintech, and green finance.

**Keywords:** finance; financial system; financial market; financial psychology; banking; Vietnam

## 1. Introduction

The COVID-19 recession is considered the worst recession ever since the Great Depression, which by far is much worse than the global financial crisis in 2009 (Gopinath 2020). According to the World Economic Outlook from International Monetary Fund in April 2020, the world GDP growth was projected to fall by 6.3 percentage points to −3 percentage within 3 months (International Monetary Fund 2020). Meanwhile, when the Global Financial Crisis 2008–2009 occurred, it was only –0.1 percentage (Gopinath 2020).

After the crisis of 2008–2009, Vietnam's GDP had soared by USD 162.791 billion, which is equivalent to the growth of approximately 264.2% in 2019 (World Bank 2021). According to the Ministry of Finance (2017), the disbursed capital of FDI in 2008 also reached USD 11.5 billion before doubling to USD 20.38 billion in 2019 (Thu Phuong 2019). The entire period witnessed a slight upward trend that ended up at USD 11.35 billion in the first 8 months of 2020 (PL 2020). Characterized by such a strong response and recovery from the effect of the 2008 financial crisis, Vietnam is thus a key emerging market worth studying. Here, it is also important to emphasize that despite a humble beginning, Vietnam is now one of the fastest-growing markets in the world only 35 years after the 1986 socioeconomic reform, Doi Moi (Locus Bulletin 2019).

Vietnam's financial market has witnessed a series of remarkable events in recent years. In particular, the VN-Index hit a new historical peak—as it reached 1252.45 points on April

2021 along with 113,875 new trading accounts opened in March 2021, which made the total number of traders in Vietnam has reach 3.02 million—equal to 2.8% of the Vietnamese population (Nguyen 2021). Due to the exceptional control of COVID-19, the trade surplus has reached a record high over 5 recent years, at USD 19.1 billion (Nguyen 2020). In 2017, the country was ranked as one of the most globalized populous countries (Kopf 2018); such status and the dynamic growth of the financial market have led to Vietnam becoming one of the most promising lands for foreign investors. Notably, in spite of the adverse effects of COVID-19, Vietnam's GDP still managed to grow by 2.9% in 2020 (Lee 2021).

Hence, examining the resiliency of the Vietnamese financial market in dealing with the outcomes of a post-crisis world after 2008, the process of international integration could offer actionable insights for other emerging markets' policy makers as well as global investors. Thus, in this paper, we set out to understand Vietnam's landscape of finance research from 2008 to 2020. Since there has been no comprehensive review of financial studies in Vietnam until now, this paper aims to contribute a systematic and critical review of key financial issues in one of the most dynamic emerging markets. Hence, it provides not only research directions but also actionable insights for researchers and policy-makers who seek to understand and strengthen financial systems in post-crisis financial markets.

The rest of the paper is structured as follows: Section 2 presents the data and methods used in the paper. Section 3 reflects on the trends of financial publication in Vietnam before reviewing in depth the four main pillars of the Vietnamese financial research, which are Structural issues, Banking system, Firm issues, and Financial psychology and behavior. Section 4 is the discussion, followed by concluding thoughts in Section 5.

## 2. Data and Methods

### 2.1. The SSHPA Database

The database we use for this review is SSHPA (Social Sciences & Humanities Peer Award), which is a nationally verified, structured, comprehensive open database specialized in social sciences and humanities research in Vietnam. This database has served to promote the transparency of science (Vuong 2020; Teixeira da Silva and Vuong 2021) and reducing the cost of doing science (Vuong 2018). All publications of Vietnamese researchers since 2008 in the fields of social sciences and humanities, which have been published in Web-of-Science and Scopus-indexed academic outlets, are tracked and recorded in the system of the database. A more detailed description of a rigorous data processing procedure and the potential for reproducibility of the system can be found in the data descriptor paper in Nature Research's *Scientific Data*, which is a top-tiered data science journal (Vuong et al. 2018b) or method article in *MethodsX* (Vuong et al. 2020c). SDA is the short form of SSHPA Data Analysis. SDA serves the function of extracting data from the SSHPA database; then, it creates an extensive collection of visualizations from these data. By applying these two stages, common problems such as duplicate data, incorrect DOI, incorrect author names, etc. will be eradicated, thus ensuring the accuracy and precision of the outcomes, which can be utilized for different research purposes.

As SSHPA is capable of high-quality data collection and clear visual display for analysis, we have extracted a subset of the financial research database to answer the following question: "What does the landscape of finance research look like in Vietnam between 2008 and 2020?" Next, the search strategy and data-filtering process will be presented.

### 2.2. Data Extraction and Filtering Process

As we aim to cover the fundamental elements of a financial market (finance, investment, stock market, banking system) and to establish an inclusive view of the research trends in Vietnamese finance, we have chosen the following set of keywords: "finance", "financial", "financing", "invest", "stock", "credit", and "bank". Entering the keywords into the SSHPA database, we found 563 publications containing at least one of those keywords in the title and/or the abstract. Then, the 563 publications were exported to an Excel sheet.

Next, we examine all the titles and abstracts to ensure there are no omitted parts caused by errors or incompatible format. Then, we further filter the most relevant papers for the Vietnamese market according to three criteria:

(1) The database is collected from Vietnam or regions in which the Vietnamese market is included or strongly influenced.
(2) The research topic of the paper is either partially related to or focused on finance in multiple fields, notably the financial market structure, financial policy, or corporate finance.
(3) The research coherently presents its results and implications for topics such as systematic corporate governance or policies implementations.

After checking each paper one by one, the selected articles are cross-checked by at least another group member. Finally, 249 articles were removed from the data pool, and the remaining 314 articles were included in the final sample.

We examine similar papers on the topic of finance research in emerging markets (Brooks and Schopohl 2018; Han et al. 2018; Linnenluecke et al. 2016) and study their analysis approach. We find that although the review papers' methodologies vary, the authors strike to identify the most significant characteristics of the finance research and analyze their trends. Within this setting, we decide to apply a systematic approach to our analysis for two main reasons. (1) The systematic approach enables us to review a large number of data with a high focus on empirical evidence and research output, which is critical for identifying the characteristics of the financial research database. (2) While the systematic approach is highly time consuming, our SSHPA database's features, which includes the displays of articles' abstracts, keywords, and significant outputs, will significantly accelerate the information gathering process.

The following figures and tables will demonstrate our final sample's characteristics. The final dataset consists of 9 conference proceedings, 7 book sections, and 226 scientific journals with JIF ranging from 0 to 4.803 (see Figure 1). In addition, Figure 2 displays a list of top publishers in the dataset. It also addresses 14 fields, many of which are very prominent for future research (Environment/Sustainability, Sociology, Healthcare, etc.) (see Table 1). With the scope of only one financial market, this dataset is relatively comprehensive compared to other systematic review papers (Brooks and Schopohl 2018; Han et al. 2018; Linnenluecke et al. 2016).

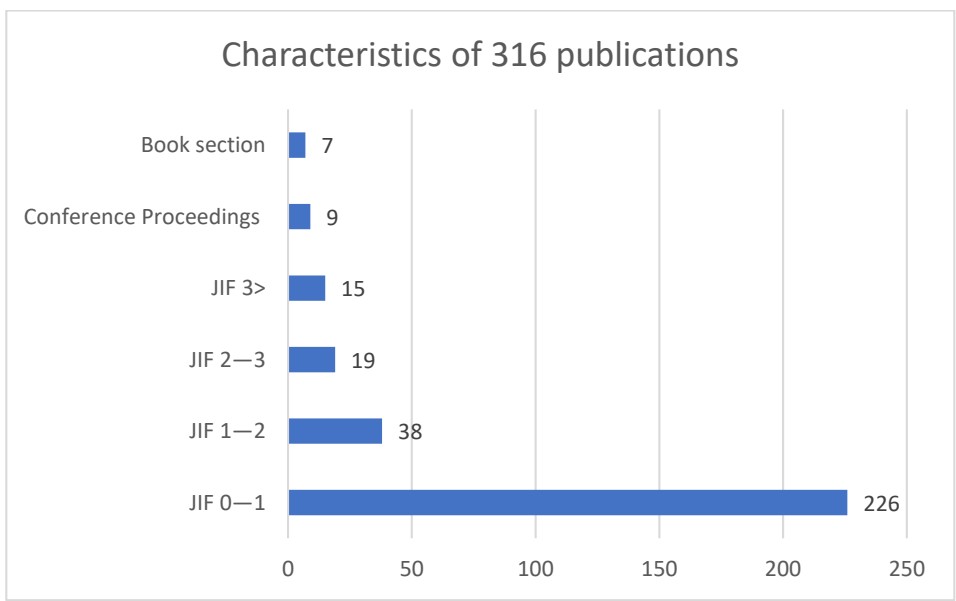

**Figure 1.** Characteristics of 314 publications from the final sample.

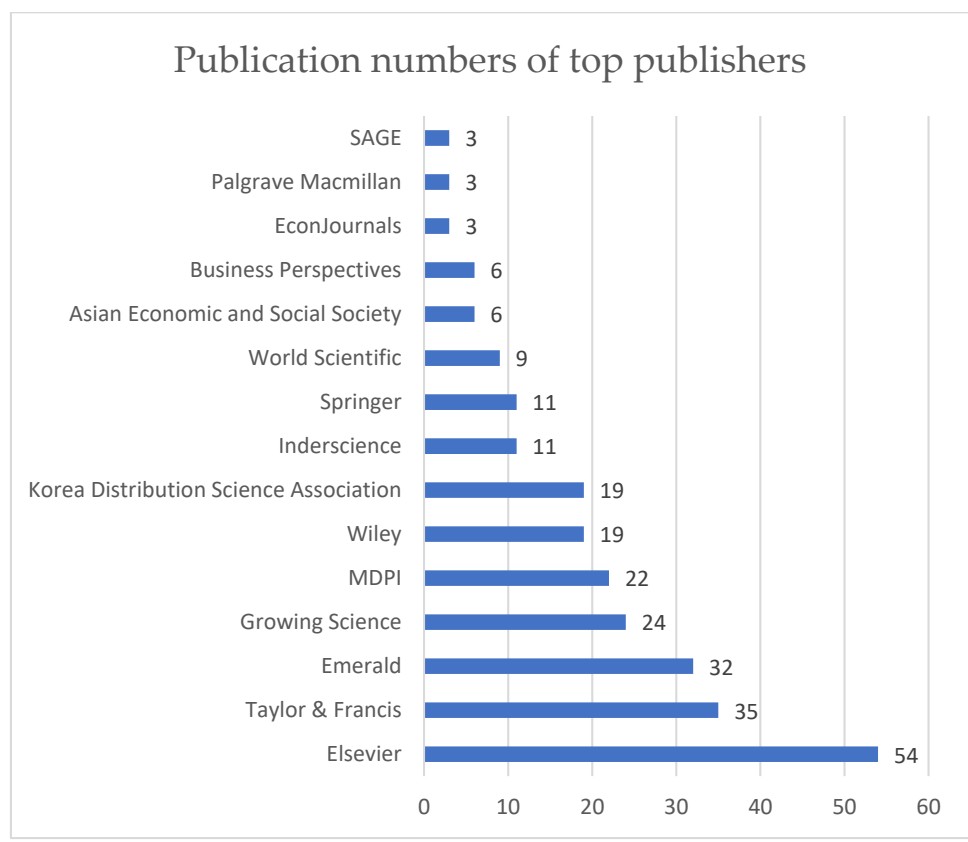

**Figure 2.** Most popular publishers by journals from the final sample.

**Table 1.** Addressed fields by publications in the final sample.

| | |
|---|---|
| 1 | Economics |
| 2 | Business |
| 3 | Management |
| 4 | Education |
| 5 | Law |
| 6 | Agriculture |
| 7 | Environment/Sustainability Science |
| 8 | Sociology |
| 9 | Political Science |
| 10 | Logistics |
| 11 | Law |
| 12 | Healthcare |
| 13 | Geography |
| 14 | Urban Studies |

## 3. Thematic Review

### 3.1. Vietnam's Financial Publication Output Trends

It is noticeable that in the period from 2008 to 2017, the number of publications in Finance was relatively small, with less than 25 articles published per year. 2018 was the first time the number of published papers surpassed the figure of 25. Then, from 2019 onwards, a sudden surge was witnessed in the Finance-related publications, with the number of papers increased nearly four times, resulting in approximately 100 papers in

each of the year 2019 and 2020, corresponding with the overall increase in the productivity of Vietnamese scientific research (Vuong 2018, 2019b) (See Figure 3). As the average annual growth rate is 42.94%, we expect a high level of publication output in the years to come.

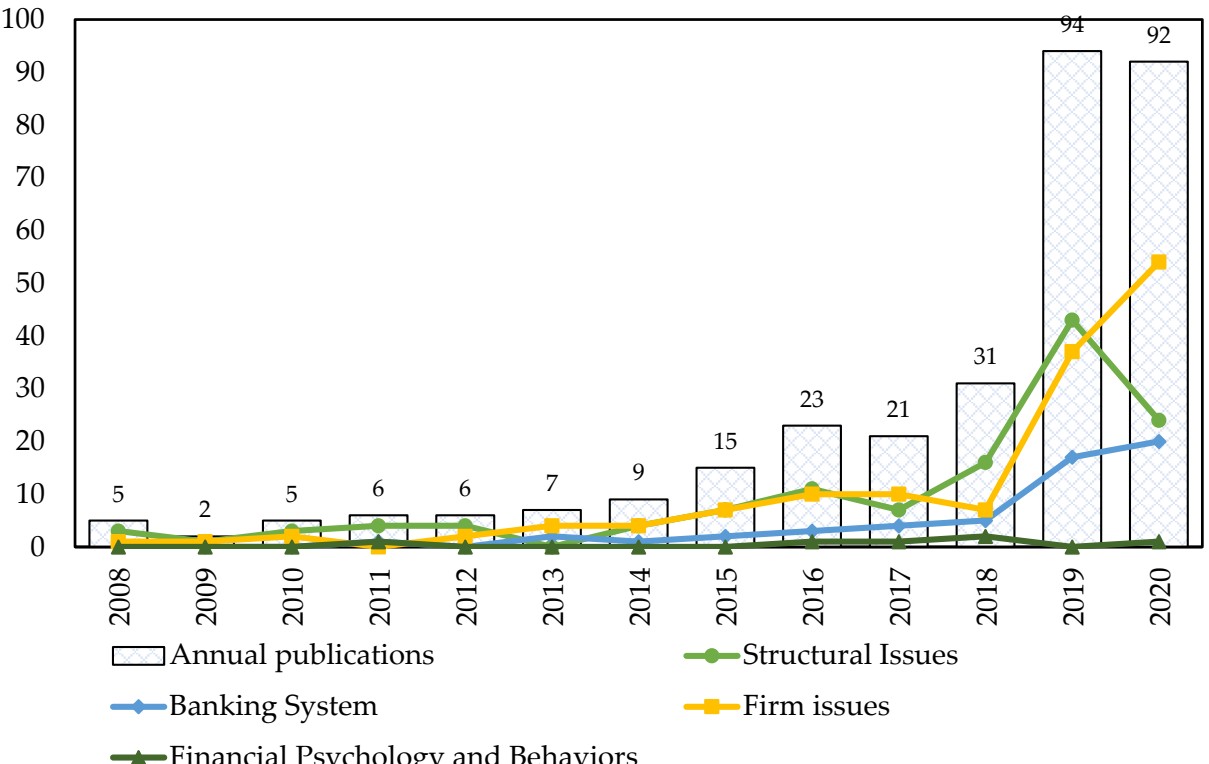

**Figure 3.** The number of published articles related to Finance in the SSHPA database, 2008–2020.

### 3.2. Vietnam's Finance Research Landscape

We categorize the articles mainly into four (4) larger themes: (1) Structural issues, (2) Banking system, (3) Firm issues, and (4) Financial psychology and behavior. The results are discussed as follows. One article may aim to provide answers for multiple problems, and thus, it could be categorized into different themes (see Figure 2).

#### 3.2.1. Structure Issues

Within this category, the research focuses on any issues that affect the financial setting from a macro perspective. As the result, the researching questions could be as broad as how prejudice affects credit assessment (Pham and Talavera 2018) or to which level we can predict a currency crisis (Pham 2017). These research studies serve the purposes of understanding the markets' mechanisms, evaluating financial policies, and establishing a more efficient financial system (Chinh and Vuong 2009a, 2009b; Nguyen et al. 2019d; Vuong and Tran 2009; Vuong et al. 2020d).

#### Stock Market

From the results and recommendations of relevant research on the stock market, we can identify researchers' two main interests: to improve financial information systems and to identify risks in the stock market.

The Vietnamese stock market is relatively new, as it was launched in 2000. While the market is constantly developing with new stocks listed annually, the challenge from information inefficiency persists over the years (Vo and Bui 2016). In other words, information on the corporates' cash flow, the book value of stock, firm sizes, etc., are either not properly listed in the firm's financial statement or not reliable or accessible (Hung et al.

2018). As information inefficiency can have adverse impacts on both investors and the firm's stock value (Vo and Bui 2016), the task of improving financial information systems is a must, and it can be accomplished via the following policies. The first is a stricter financial statement regulation from the State to enforce compliance from listed companies; as Hung et al. (2018) points out, complete disclosure of financial information is also encouraged for firms to gain trust from investors. Second, a healthy increase of foreign ownership could improve the informational efficiency of the equity market (Vo 2017a), as stock price informativeness, i.e., how changes in earnings are presented in the stock price (Hu et al. 2018), is found to fluctuate in accordance with the increase in foreign ownership. A robust financial information system is also critical to carry further research on the stock market, which is reflected in stock returns and stock liquidity studies (Dang et al. 2019, Huynh et al. 2018, Pham et al. 2018a).

Multiple studies point out that the Vietnam stock market is vulnerable to the strong influence of foreign markets and foreign investors. The VN Index is shown to be influenced by foreign stock markets' fluctuation; as Vo (2020) shows, an extreme surge on stock markets in Taiwan, China, Japan, Hong Kong, Singapore, and Korea could create a similar spillover effect to the Vietnamese market. Nguyen (2011) has identified the spillover effects of US macroeconomic news on the Vietnam stock market. Another research studies the speculative stock market's bubble formation in Asia and Latin America to apply to the emerging stock markets (Tran 2017). Stock splits, while common for many firms, are found to be correlated with vulnerable firms and insider trading (Nguyen et al. 2017b). Furthermore, foreign investors have a high potential of destabilizing stock prices if they react to information simultaneously, as a time-series analysis of foreign investors' trading on stock returns in Vietnam reveals their sophisticated timing strategy: they are long-term and positive feedback traders; i.e., they hold their stock for a long period and only buy when the stock price has decreased and sell when stock price has increased (Vo 2017b). The sensitivity of the Vietnam stock market to changes in foreign markets and foreign investors highlights the importance of (i) finding a countermeasure for herd mentality (Faber and Vuong 2004; Dang and Lin 2016) and (ii) the early detection of risks from overseas markets and investors (Batten and Vo 2015; Vo 2017a, 2017b, 2020). The influence of foreign ownership will be further discussed in Banking and Firm sectors, while herd mentality will be explained in Financial psychology and behavior.

Foreign Investment

Over the past two decades, research on FDI in Vietnam has converged on two main areas of interest. First, it concerns how to generate a virtuous feedback loop between economic growth and FDI attractiveness. An early research on FDI by Anwar and Nguyen (2010) has pointed out the positive correlation between Foreign Direct Investment (FDI) and economic growth (Dinh et al. 2019). Economic growth was determined through human capital, education and training level, financial market development status, and technology development (Anwar and Nguyen 2010). However, recent studies show that to further enhance the FDI absorptive capacity, improving these aspects is necessary (Dinh et al. 2019).

Second, this area of research focuses on limiting the negative impacts of FDI. For example, although FDI has positive impacts on multiple sectors, notably imports–exports, FDI could pose a threat of macroeconomic instability (Anwar and Nguyen 2011; Le 2016). Meanwhile, during the Asian financial crisis, FDI is found to be vital in boosting net exports (Anwar and Nguyen 2011); Le (2016) finds booming FDI real estate projects lead to the risk of macroeconomic instability in causing real estate inflation, destabilizing capital balance, and increasing bad debt ratio in the commercial banking system. Notably, FDI could exacerbate regional income inequality, as the positive correlation of FDI and economic growth is only identified in the Red River Delta, North East, South East, and Mekong River Delta, according to Anwar and Nguyen (2010). Thus, policies and research studies to understand how to promote FDI in the remaining unattractive regions are essential (Le

2016). Future studies can focus on the impacts of regulations on the construction of these FDI projects, and how to engender a shift of FDI flows from real estate to manufacturing sectors as well as how to increase eco-friendly FDI (Vuong 2021). This is to create a more sustainable financial market and distribute the economic gains of increased FDI equitably.

Financial Inclusion Promotion

Research on Financial inclusion is relatively new, being published mainly in the 2019–2020 period. This may relate to the ratification of the National Financial Inclusion Strategy in 2020, which will continue until 2025 with a vision of 2030 (Vietnam Government Portal 2020). To prepare for the establishment of a nationwide financial network, policymakers and researchers are studying financial inclusion using global datasets (Le et al. 2019a; Morgan and Long 2020; Van et al. 2019; Vo et al. 2019).

In simple terms, financial inclusion is the ideal accessibility of every demographic to beneficial and affordable financial products or services (Van et al. 2019; Vo et al. 2019). Financial inclusion has been widely researched for its potential to mitigate poverty and boost economic growth, to positively impact income quality, macroeconomic stability, and inflation control (Le et al. 2019a; Van et al. 2019; Vo et al. 2019). On the other hand, there is evidence for the adverse effect of financial inclusion and financial sustainability on financial efficiency (Le et al. 2019a). Several proposals have been put forward to increase financial inclusion while limiting its side effects. At the grassroots level, one solution is educational programs that improve general financial literacy (Morgan and Long 2020). More importantly, bank competition also seems to be encouraging financial inclusion (Pham et al. 2018b). In the future, we can expect a growth of research studies on the implementation outcomes of Vietnam's National Financial Inclusion Strategy in Vietnam.

Rural Credit and Microcredit

In emerging markets, rural credit and microcredit play a significant role in promoting credit accessibility to the impoverished and low-income demographic. As Vietnam's agricultural sector has grown substantively over the past 35 years, microcredit programs, offered by the formal and semiformal credit channels, have grown in demands and complexities, notably in loans' size and the repayment period (Khoi and Gan 2016; Nguyen et al. 2019d). The effectiveness of such microcredit programs has been a popular topic.

On the one hand, the microcredit programs have been found to successfully address the poor households at the bottom of the low-income demographic, positively increase the probability of child education, and improve rural household's self-employment profits, food consumption, and social network (Haughton and Khandker 2016; Lensink and Pham 2011; Phan et al. 2019, 2020a; Poon and Thai 2010; Tran et al. 2019b). On the other hand, there have been concerns about whether the targeting of such projects is pro-poor. Examining the preferential credit program from VBSP in 2002 and 2004, the participation rate of impoverished households in rural areas was only 12% and 6.4% nation-wide. The non-poor households also comprised approximately 67.1% of all participants and received more credit than the impoverished households (Cuong 2008). Furthermore, when examining the positive impacts of microcredits closely, there are flawed indexes. For instance, if households' incomes stay below minimum wages, microcredit's impact on child education probability becomes negative (Do 2015; Tran et al. 2019b). This highlights the importance of other measures to increase microcredit's role in alleviating poverty such as risk coping measures and farming technologies.

Research on Rural Credit and Microfinance proves the importance of credit accessibility in poverty reduction in rural areas. Credits open opportunities to develop businesses, the same foundation on which a lot of Small and Medium Enterprises (SMEs) and Startups are emerging. It is also important to note as climate change disproportionately affects small local businesses; future studies can focus on the possibility of financial products that

alleviate such adverse impacts (Vuong 2021). We will further discuss the influences of credit in the Firm issues section SMEs Financing.

IFRS Adoption

Vietnam is one of the countries where International Financial Reporting Standards (IFRS) are not compulsory for financial statements. Researchers are examining the possibility of IFRS adoption by examining financial experts' view on IFRS or its compatibility to Vietnam institutional context (Le et al. 2019b, 2019c, 2020b; Nguyen and Rahman 2019; Phan et al. 2018; Phan 2014; Phan and Mascitelli 2014).

The current Vietnamese Accounting Standards (VAS) are considered to be limited, specifically when dealing with recognition of the fair value and incurred financial tools (Nguyen and Rahman 2019). Meanwhile, the widespread adoption of IFRS could improve Vietnam's FDI attractiveness (Gordon et al. 2012). For example, according to the experts, IFRS adoption will promote the transparency of financial information, decrease misstatements in financial reports, and increase the level of responsibility for firms to disclose information (Le et al. 2020b). However, high-cost expenditures of IFRS preparation, language barriers, and legal framework issues are the main barriers of IFRS adoption (Le et al. 2019b, 2019c, 2020b). Although the alignment of Vietnamese accounting legislation with IFRS was officially announced in November 2013, the detailed directions of IFRS adoption have not been cleared (Phan 2014).

Similar to Vietnam's approach to economic reform (Vuong 2019a; Chinh and Vuong 2009a), the institutionalization of IFRS in Vietnam happens gradually, following a legalism-based accounting regulation approach (Nguyen and Rahman 2019). Here, the state initiates the IFRS-oriented rules to local enterprises and businesses, which would cause reassessment on exiting accounting statements. Grassroots criticisms after the "test" will help the state to adjust IFRS to suit local needs. Despite this state-led process, Big4 firms and Western professional accounting associations (WPAA) have contributed greatly to the institutionalization (Nguyen and Rahman 2019). Experts' estimate that in order to efficiently integrate IFRS into firms' financial statements, the optimal time for IFRS preparation is around 3–5 years, yet it has not materialized until now (Le et al. 2020b; Phan 2014; Phan and Mascitelli 2014). Vietnam would need to encourage enterprises to voluntarily apply for IFRS and then gradually set the compulsory standard for financial statements, while enterprises need to support human resources training to increase IFRS knowledge for the local accountants (Le et al. 2019c).

3.2.2. Banking System

After sorting, 72 articles discuss numerous banking issues. Specifically, the three topics of risks, performance, and policies garnered the most research interests.

Risks

Regarding bank risks, Vietnamese researchers focus on examining factors affecting bank risks and their preventive measures. Interestingly, empirical results from Vietnam reveal significant inconsistencies in the literature concerning bank risks.

Regarding the effects of banks' income diversification, the traditional view states it would reduce bank risks, which has been demonstrated in studies of various markets: the EU (Saghi-Zedek 2016), Japan (Sawada 2013), and a sample of 111 countries (Doumpos et al. 2016). Nevertheless, Batten and Vo (2016)'s study shows that in emerging markets such as Vietnam, banks with a higher income diversification face higher levels of risk. This result was even more robust when different measures of diversification, including fee income and trading diversification indicators, were used to test this relationship. Such empirical results are they are in line with Stiroh and Rumble (2006) in the US, revealing an inconsistency in the literature.

In Vietnam, bank credit risk seems to positively correlate with bank size, contradicting results from other emerging markets (Kasman and Kasman 2016). Batten and Vo (2016)

found that banks with higher expenses were associated with higher risks (Batten and Vo 2016). Concurrently, Le et al. (2020a) found that small banks seem to be more efficient than large banks as their input waste, output shortage, and risk surplus were just one-third of those of large banks. Large and well-capitalized banks in Vietnam were also proven to be less likely to increase credit without taking risks into account; in fact, after the crisis, they were better at considering the risks by reducing their credit supply (Vo and Nguyen 2014). If a tight monetary policy had existed, large banks would have faced more bank risk (Vo and Nguyen 2014).

Hoang et al. (2019a) found that banks' credit risk, measured by the NPL ratio, was negatively affected by the unemployment rate and the real interest rate, while there was a significant positive association between loan loss reserves and the credit risk. These findings were different from the results of Boudriga et al. (2009)'s study, which applies the same FEM-REM estimation, indicating that a lower loan losses reserves ratio would increase the NPL next year. Since the loan loss provision ratio affected positively the credit risk, the authors suggested the State Bank should supervise frequently and rigorously commercial banks to ensure compliance with regulations, such as Circular 02/2013/TT-NHNN, which states commercial banks need to restructure, classify impaired loans, and make sufficient provision for each group of uncollected; or Resolution No. 42 on handling bad debt in credit institutions (Hoang et al. 2019a). Between 2008 and 2010, as a result of the global crisis, Vietnam's credit risk increased significantly, as the growth rate of non-performing loan (NPL) reached 51%, twice as much as this period's average credit growth (Le 2017). The NPL ratio in Vietnam peaked at 4.86% of the total outstanding loans in 2012 before going down to 2.55% in 2015 and remained stable since then, proving a relative success of post-crisis commercial banks regulation (Le 2017; Hoang et al. 2019a).

Using non-parametric and Copula approaches, Huynh et al. (2020) found stock price information reflected that a bank's risks may transfer to other banks through stock returns, confirming the existence of a contagion risk. Consequently, Huynh et al. (2020) provided crucial insights into the dynamics of contagion risk, which are highly informative for policy-makers as well as investors to contain those risks:

1. A bank holding public capital in Vietnam will positively affect the other banks.
2. Contagion risk in the banking system emerges when banks are cross-owned, such as Vietnam Commercial Joint Stock Export-Import Bank A Chau Bank (ACB) and Joint Stock Commercial Bank for Investment and Development of Vietnam (BID).
3. A commercial bank with state-owned capital may lead to contagion risk if they operate inefficiently or invest in a weak bank.
4. Banks with a high ratio of NPL can cause a high contagion risk.

Bank Structure

Bank structure—the study of banks' organization system—is the second-most appealing topic to Vietnamese researchers in the banking sector. Crucially, "Bank restructuring"—financial reform—was a key area of concern. Around the world, a huge volume of studies has discussed the interrelationships between bank structure and bank efficiency. Some, for example, Berg et al. (1992) and Zaim (1995), claimed that bank performance was better after deregulation, while the banking efficiency of large banks in the US was proven to remain the same (Elyasiani and Mehdian 1995). In Vietnam, there have been three major bank restructuring initiatives, corresponding to the post-Asian financial crisis (1998–2003), the WTO joining (2005–2008), and the restructuring of the Vietnamese economy (2011–2015).

Here, bank efficiency was found to decline during the bank structuring processes, as Vo and Nguyen (2018) showed the different restructuring schemes including the privatization of state-owned commercial banks, state intervention, and mergers and acquisitions (M&As) did not improve efficiency in the long term. Additionally, Vietnam's M&As activities do harm to the return on assets (ROA) while its effects on return on equity (ROE) and net interest margins (NIMs) were not clear (Nguyen and Tran 2019). These results were the inverse of Fatima and Shehzad (2014)'s, as they concluded that M&A activities had an

impact on ROE, while the influence on ROA was not clear. The role of the government is crucial, as the Vietnam government's restructuring policies in the first stage (1998–2003) had not yet supported the banks to implement the restructuring (Vo and Nguyen 2018).

On the other hand, liquidity creation, which may increases banks' insolvency, may be decreased by the application of Basel III (the third part of a voluntary global regulatory framework for banks' capital adequacy, tension testing, and market liquidity risks, which were developed to cope with deficiencies in financial regulation revealed by the 2007–2008 financial crisis) (Le 2019), and banks being involved in more non-traditional activities for fees and commissions tends to reduce this component (Dang 2020). Le (2019) also pointed out that large banks were the strongest contributor to the growth of liquidity creation (LC) in general, and a negative two-way relationship between LC and bank capital in Vietnam was evidenced from 2008 to 2015. In terms of a bank's internal control, Dinh and Tran (2019) found that the Control Environment affected the efficiency of internal control the most, followed by Information and Communication and Credit Status Ranking. The variable that had the least impact on the Effectiveness of Internal Control was Control Activities.

These results highlight the importance of communication and cooperation between the regulators and the banking sector to effectively improve bank efficiency during the restructuring process.

Banking Performance

Vietnamese researchers assessed bank performance by various criteria and some of which were different from the world. For instance, regarding the measurements for bank performance, there were plenty of criteria among researchers worldwide: interest margins (defined as the difference between interest income and expenses divided by total assets) was used by Revell (1982) for US commercial banks, or Kosmidou and Zopounidis (2008) selected a variety of particular ratios such as Net Income before taxes/Equity, Loans/Deposits, Equity/Total Assets, etc. for the evaluation of commercial banks in Greece. Meanwhile, even though when it came to measuring bank performance, profitability ratios such as ROA and ROE were commonly and traditionally used due to their advantages (Lee and Kim 2013), it was surprising that Vietnamese researchers used these two aspects in only a small number of studies. Luu et al. (2019) used the average assets (ROA) and return on equity (ROE) as the two proxies for bank performance, whereas Nguyen (2019) combined ROA and ROE with the bank risk (Z-score) to judge the bank performance in his article. The non-radial slack-based directional technology distance function (developed by Färe and Grosskopf (2010)) was the tool of Le et al. (2020a) to analyze the risk-adjusted efficiency, and the bank performance was also assessed via the cost efficiency of the banks through the data envelopment analysis (DEA) and the stochastic frontier analysis (SFA) (Nguyen and Pham 2020) or the NIMs as what Nguyen et al. (2020b) conducted. The SFA method was also used to study alternative treatment for NPLs in bank cost efficiency studies (Ngo and Tripe 2017). In the meantime, credit risk, bank profitability, and bank solvency were the three issues that Dang (2019) has taken into account when it came to reviewing the bank performance.

Vietnamese researchers focused on studying the correlations between bank performance and particular components such as excess liquidity, loan growth, income diversification, etc. Speaking of negative effects, bank performance was said to be affected by the loan growth indicators (Dang 2019), excess liquidity (Nguyen et al. 2020b), bank size (Vo et al. 2019), and revenue diversification (Nguyen 2019). According to Dang (2019), the loan loss provisions would be increased from 2 to 3 subsequent years by the growth in lending and would also decrease the bank capital ratio in the next year. Even though the bank profitability was beneficial from loan growth both in the short term and long term, the banks' expanding lending activities aggressively tended to lower the bank solvency immediately.

Meanwhile, Nguyen et al. (2020b) have proven that the excess liquidity would compress the NIMs, which reduced the efficiency of policy interest rates on NIMs, as higher

monetary policy interest rates would expand NIMs. This indicated that the excess liquidity showed a tendency to induce banks to reduce lending interest rates to expand credit supply. Regarding the bad effects of revenue diversification on bank performance, Nguyen (2019) concluded that profitability is negatively impacted by diversification. Nonetheless, the more diversified the listed banks, the more increased the bank's stability (Nguyen 2019). The author also suggested that "the Vietnamese commercial banking system needs to improve and enhance the non-credit service quality, especially e-banking services, to meet the trend of competition in banking digitization and the trend of consumer consumption. The better the banking liquidity is, the higher the profitability of the credit institutions." (Nguyen 2019). These findings were quite in contrast to what foreign researchers found, as they claimed that banks in emerging countries could benefit from revenue diversification (Sanya and Wolfe 2010; Nisar et al. 2018).

On the other hand, the bank performance was beneficial from the income diversification (Luu et al. 2019), bank size, ownership, and age (Vu et al. 2019). However, Batten and Vo (2019)'s study found that bank size would have a negative impact on bank performance; on the other hand, bank capital and bank productivity were proved to be advantageous. Luu et al. (2019) also did an additional analysis to assess whether bank experience and ownership structure affect the interrelationship between bank performance and income diversification and found that the diversification would have a positive impact on state-owned and foreign banks, while for non-state-owned domestic banks, bank performance appeared to be lowered if banks increasingly diversified towards non-traditional businesses. This was quite similar to what Vu et al. (2019) found: not only were state-owned banks more efficient, but larger size and longer time in servicing also had a positive impact on bank performance.

Bank Policies

Speaking of the shifts in the Vietnamese bank's lending rate, policy-related rate, and monetary policy post-crisis, Nguyen (2015a) aimed at answering three key questions. The first one was whether there are asymmetries in the Vietnamese lending policy-related rate spread and, if such asymmetries existed, how lending and central bank's policy-related rates responded to these asymmetries, and whether responses to such asymmetries were independent or were dynamically interrelated (Nguyen 2015a). The author has found that the asymmetries existed, and the spread adapted to the threshold faster when the central bank's policy-related rates fell in relevance to the lending rates than when the central bank's policy-related rates followed the opposite trend (Nguyen 2015a). Moreover, it was evidenced that there was a bidirectional Granger causality between the Vietnamese lending rate and the central bank's policy-related rate, and these rates affected each other's movements (Nguyen 2015a). The second question is whether the Vietnamese lending institutions exhibited competitive or predatory pricing behaviors, and to what extent (Nguyen 2015a). It was indicated that the competitive pricing behavior existed, but it contradicted those studies on the rate-setting behaviors of lending institutions since the predatory behavior was witnessed in Russia (Nguyen et al. 2017a) and Thailand (Nguyen 2017a). The last one was whether the variance of the basis from one month impacted the variances and spread in the subsequent months (Nguyen 2015a).

In the meantime, Pham and Lensink (2008) chose ACB bank lending to small and medium-sized enterprises (SMEs) in Vietnam as a case to study the determinants of loan contracts to business firms in Vietnam. Loan contracting included the lender–borrower relationship and concentrated on banking, skipping other issues related to the financial structure of the firm (Pham and Lensink 2008). In particular, the authors would examine the three main loan contract features that banks use in lending to business firms, which were loan maturity, collateral, and loan interest rate. The results have shown that strong interdependencies between these contract terms with significant bidirectional relationships between collateral and loan maturity, loan rate and loan maturity, and a unidirectional relationship between the loan rate and collateral existed. Each interrelationship mentioned

above was aligned with particularly well-known theories of financial contracting such as those of Merton (1974); Bester (1985); Arnoud et al. (1991); or Diamond (1991). The authors also looked forward to furthering research focusing on the theoretical aspect of this preliminary proposition to study the conditions, and to what extent, bank behavior and borrower behavior determined outcomes (Pham and Lensink 2008).

3.2.3. Firm Issues

In this section, 125 articles discussing various issues related to Vietnamese firms are examined. The results present some of the salient issues of Vietnamese firms, which are the topics of capital structure, corporate governance, ownership, and SMEs financing.

Capital Structure

Capital structure is one of the vital factors affecting firm performance, and it can be defined as the ratio of debt and the ratio of equity to the total capital of a firm (Nguyen and Le 2017). In terms of capital structures' relations, Vo and Ellis (2017) argued that there is a negative relationship between financial leverage and shareholder value. Moreover, Hoang et al. (2019b) concluded that capital structure negatively relates to financial performance. Nguyen and Nguyen (2020a) confirmed this finding in their study and further stressed that state-owned enterprises were more strongly influenced by capital structure than non-state enterprises. This result is comparable to the market of India and China, where state-owned enterprises performed the worst, and government ownership is influenced by political interference. Hence, Nguyen and Nguyen (2020b) suggested some recommendations for better-utilizing capital structure. On the firm level, business managers are recommended to establish proper measures. These measures should not only aim to use debt and increase sales but also ensure the improvement in business performance. On the state level, the government is advised to adjust the interest rates at a reasonable level to assist businesses in achieving efficiency (Nguyen and Nguyen 2020b).

Corporate Governance

Corporate governance is a system designed to protect investors' interests. More than that, it acts as a tool directing a company to a balance status, which improves its health and its brand health (Tuan 2014). This section will elaborate on the role of the board of directors, and the definition of corporate social responsibility and its application in the Vietnamese setting.

The board of directors is a determinant of corporate governance. Nguyen and Dang (2020) and Dang et al. (2020) pointed out that the board of directors directly influences the firm value and earning quality. In the context of the Vietnamese economy, where enterprises are mostly small- and medium-sized, the expansion of the board of directors should be compatible with the company size. However, the expansion should be moderated, since a bigger board of directors may lead to extra administrative costs and contradictions between members. Therefore, an efficient managing board must be one consisting of competent and experienced members (Dang et al. 2020; Nguyen and Dang 2020).

An aspect that cannot be ignored in corporate governance is corporate social responsibility (CSR), which is a strategy devised to respond to new societal requirements (Nguyen et al. 2019a). According to Kabir and Thai (2017), firms that are more active in CSR activities tend to have better financial performance (Kabir and Thai 2017). However, CSR may be used in unnecessary projects, or it can be exploited to polish the names of opportunistic managers (Nguyen and Trinh 2020). Therefore, CSR has to be utilized with good morals (Le et al. 2019d; Nguyen and Ngoc 2020; Nguyen et al. 2019a).

Specifically, in the setting of Vietnam, though frequently being mentioned, CSR has not yet received deserving attention. Vietnamese firms still exhibit a neglected attitude toward environmental issues, and the country is lacking a platform where environmental news can be delivered to the public accurately and transparently. Therefore, it would be ideal if the government could set up a public database that holds records all of the

environmental incidents caused by companies as well as keeps track of the promotions of environmental initiatives, reports on businesses' engagement in sustainability, or long-form writings about the environment. The establishment of such a platform would create a positive impact in raising the attention and awareness of the public and businesses toward the environment as well as set an example for other countries to follow (Vuong et al. 2020b).

Ownership

The relationship between ownership concentration and firm performance has been a regularly researched topic for over two decades. Ownership concentration is an important mechanism of corporate governance to restrict the agency problem from the separation of ownership and control (Nguyen et al. 2015). In a developing market such as Vietnam, owner concentration is significantly and positively related to firm performance (Tran and Le 2020). Nguyen et al. (2015) also attributed the positive relationship to the highly constructed ownership structure. They also pointed out that the better the governance from the national government, the more positive the performance of firms. Thus, more effort should be put into enhancing national institutional characteristics, such as investor protection or rule of law, which in turn will facilitate a better environment for businesses and stakeholders (Nguyen et al. 2015).

With the flow of funds increasingly pouring into the emerging market of Vietnam, the topic of foreign ownership is also widely mentioned in recent years. Multiple research studies have shown that foreign ownership is associated with less liquidity but higher corporate cash holdings, and it is related negatively to corporate investment efficiency (Tran 2020; Vo 2016, 2017b). These findings lead the way to further implications for the policymakers. First, the government should establish policies to attract foreign investors to invest in firms in a long-term period. This will reduce the volatility of stocks as foreign traders could not trade a stock very often, and more importantly, it will increase firm performance, since foreign investors will bring managerial experience, technology transfer, clients, etc., to serve their investment in the long run. Secondly, the government should focus on creating a transparent and stable economy, which would earn the trust of foreign investors. In return, international investors would prefer to invest in high transparency and a less uncertain economy for their investment projects (Nguyen and Phan 2019; Tran 2020; Vo 2016; Vu et al. 2019).

In addition to foreign ownership corporations, state-owned firms, which tend to be in strategic industries in a socialist-oriented economy (Nguyen et al. 2020c), also attract public attention to their performances (Tran et al. 2019a). In comparison between state-owned firms and privately owned firms, although the empirical result is mixed in terms of financial performance, Tran et al. (2019a) concluded that state-owned firms have better earning quality in the Vietnamese market. The reason is a worrying trend of manipulating earnings from private firms to upgrade their positions, which is a practice that is similar to the Chinese market (Tran et al. 2019a).

SMEs Financing

In emerging markets, Small and Medium Enterprises (SMEs) hold a substantial role as the dominant contributors to economic development. The three main interests of SMEs research are credit sources, credit accessibility, and SMEs' performances under credit constraints.

For the credit sources, networking with government officials and political connections are important for SMEs to obtain credits through formal channels (Archer 2019; Cao 2019; Nguyen et al. 2020a). Nevertheless, the volume of bank lending among those who successfully obtain credits shows no correlation, which suggests that official networks are necessary only for the initial loaning process. Those who have successfully obtained formal credits are not bound by governmental connection (Cao 2019). Interestingly, further extension of social networks reduces the necessity for credit loans (Le and Nguyen 2009). As the procedures to obtain formal institutional credits prove to be a barrier for SMEs,

social networks act as tools for seeking alternative informal credit sources, despite their unreliability and insufficient quantity (Cao 2019).

Entrepreneurs denied short-term loans are more likely to apply for credits from private money lenders, while those denied long-term loans prefer trade credit (Le and Nguyen 2009). Education levels also seem to be a factor contributing to credit source rationalizing, as entrepreneurs with low-level education are likely to be discouraged by debt aversion and burdensome paperwork of the formal credit channels. Thus, adjustments in the formal credit assessment system to simplify the procedures are needed, combining with government assists to improve entrepreneur education (Nguyen et al. 2019b). By improving the credit assessment environment, regional governments can indirectly encourage SMEs to take the initiative in choosing formal credit channels.

In Vietnam, the lack of capital is arguably the biggest impediment of SMEs (Nguyen 2015b). Therefore, it is vital to identify the determinants of credit accessibility, thus supporting SMEs to have easier access to capital. Owner characteristics, particularly educational level and gender, are the most important factors in deciding access to credit. Other following factors are the SME's relationship with banks and customers, collateral, sources of credit, and technology qualification. Especially, with similar characteristics, in comparison to SMEs located in the north-central area and central coastal area, the probability of credit constraints in Red River Delta and south-eastern region increases, while it decreases with SMEs located in Northern Midlands and Mountainous (Ha et al. 2016; Nguyen 2015b). These results yield several recommendations to improve the credit accessibility of SMEs. Firstly, any policy targeting to improve SMEs credit accessibility should pay more attention to small-sized and female-owned enterprises, since these groups tend to less access to credit. Secondly, to encourage credit accessibility for SMEs, the collaboration between SMEs and banks is important to diversify collateral, increase collateral value, and reduce interest rates.

Thirdly, changes should be originated from enterprises themselves. Reducing the extent of external debt dependency and using acceptable sources for various purposes are effective ways to reduce interest rate risk. They should also consider employing a commercial credit policy and aim to increase revenue by using a discount policy with the required discount rate. Last but not least, management capacity and ability need to be upgraded to access information, especially business-critical information (Ha et al. 2016; Nguyen 2015b).

Credit constraint is a persistent barrier to the performance of SMEs. Credit constraint is defined as a restriction or limitation on individuals, families, and businesses borrowing from the formal financial sector. In Vietnam, results from studies show that rural areas and male-owned firms are more constrained when accessing credit. Private firms, limited, or share-holding companies are more constrained on credit than household enterprises. In addition, younger owners tend to face credit constraints more frequently than older ones. Generally, larger firms might be less constrained, while constrained firms are less likely to expand (Nguyen et al. 2019c). Furthermore, in correlation with innovation, unlike findings from developed countries proving that credit constraints negatively affect firms' innovation, Archer et al. (2020) show that credit constraints do not always hinder the innovation of Vietnamese SMEs. Thus, to enhance innovation, managers should produce more entrepreneurial strategies to maintain their firm's value and survival. More importantly, to reduce the credit constraint of SMEs, the policymakers should seek to reform the banking sector to facilitate better credit access for SMEs (Archer et al. 2020). Or, the policymakers in Vietnam should also consider partnering with credit providers to better handle SMEs' demands, with measures such as diversifying credit packages by duration and interest rates to fit firm demands (Nguyen et al. 2019c).

3.2.4. Financial Psychology and Behaviors

The psychology of firm managers and investors in the Vietnamese market is influenced by many factors such as financial constraints, herd psychology, and foreign investors' actions.

Vuong et al. (2016) confirmed that preparedness, financial resources, and participation in social networks are critical factors influencing entrepreneurial decisions. Entrepreneurs with financial constraints would only go forward with a business idea when the probability of success is high. Otherwise, they would procrastinate to wait for more favorable conditions despite the vagueness of "favorable". However, preparedness could trump this effect as shown by Vuong et al. (2016), whether financially constrained or unconstrained, entrepreneurs who already formed a business plan are likely to launch their businesses sooner (Vuong et al. 2016; Vuong 2016a, 2016b).

Financial conditions not only affect entrepreneurial decisions but also investment behaviors, a study by Tran and Le (2017) shows. Financial conditions can be referred to as the current state of financial variables that constitute the supply or demand of financial instruments applicable to economic activity. Only investment behaviors of firms with negative cash flows are influenced by financial conditions, in the sense that improved financial conditions reduce the amount of "negative" financing constraints (i.e., the sensitivity of investment to negative cash flow). In addition, the magnitude of such impact is determined by the firm size and ownership, as this impact is stronger for larger companies, and it is more pronounced for firms that are not state-owned (Tran and Le 2017).

In terms of debt-related behaviors, in Vietnam, companies seem to prefer to reduce overall debt to raise investment. On the other hand, the long-term debt ratio does not affect firm investment as significantly as in the case of UK firms (Phan 2018). Thus, the government should establish a favorable environment to support firms access debt from various channels, including banks, equity market, and capital market (Phan 2018).

To diverge, herd behavior is also considered to be a powerful factor affecting financial players. Herding behavior is induced due to some micro-structure characteristics of the Vietnam market such as lack of transparency in information and financial management, a high degree of market volatility, etc. (Tran and Truong 2011). Several studies of various periods using different models point out the existence of significant market-wide herding behavior in the Vietnamese stock market (Faber and Vuong 2004; Kallinterakis 2007; Tran and Truong 2011). Here, international investors' engagement could mitigate such behaviors as it could minimize impulsive spending among local investors and help them strive for more professional buying strategies. Particularly, local investors can recognize future investment opportunities based on foreign investors' stock-holding decisions (Bekaert et al. 2017).

The psychological aspect is a topic on which more research attention should be focused. As the Vietnamese market possesses many cultural peculiarities as well as fast-evolving institutions (Vuong 2019a), the flow of financial information is difficult to manage. Thus, theory like mindsponge and its application can be useful for future research direction (Vuong and Napier 2015; Nguyen et al. 2021b). The application of these dynamic theory can help generating a kind of natural experiment to test various theories on financial psychology and behavior. As such, the number of current papers on this topic is underwhelming compared to its potential.

## 4. Discussion

This study suffers from a number of limitations. First of all, SSHPA is a database dedicated to Vietnamese researchers' publication. Thus, foreign authors' work on the Vietnam financial market might not be accounted for in the database. Secondly, the thematic results might be partially subjective as we have observed some of the major sections overlap with the keywords' content, particularly with "stock", "credit", and "bank". Finally, we recognize the potential bias of the systematic approach, in which the narrative is reshaped through selective empirical evidence.

However, through the detailed analyses, we have come to understand the distinctive characteristics and focuses of each topic in finance research in Vietnam. There is no doubt that finance research is continuously contributing to the development of emerging financial markets in general and Vietnam's young financial market in particular. Our review enables us to identify the following three fertile areas for future research.

### 4.1. Continued Interests in International Integration Post-COVID-19

Vietnam's integration into the international market is one of the most common research topics. The adoption of international financial standards such as IFRS and the establishment of policies for firms with foreign ownership are two of many examples of Vietnam strengthening its financial system and legal framework to connect extensively with foreign markets (Nguyen and Rahman 2019; Phan et al. 2018; Vo 2017a).

As COVID-19 ravaged the global economy and a deep recession is inevitable (World Bank 2020), Vietnam's post-COVID-19 monetary regulations and foreign trades would gather much research interests. Moreover, since Vietnam has done an outstanding job in controlling the COVID-19 outbreak, thus securing a safe and secured destination for foreign investment, changes and impacts of FDI on the domestic market will be a fertile research area.

### 4.2. Financial Technology Research

Fintech has started to become a fast-growing sector in Vietnam with the explosion in popularity of internet banking services. Notably, there are very few studies regarding the development of Fintech in Vietnam. Future finance research focusing on the access of online financial services, the efficiency of financial technology systems, or social impacts and regulations of this disruptive technology is expected to grow. Aside from the technology integration aspects, Fintech research may further inspect the influence of Vietnam's culture on entrepreneurs' willingness to adopt Fintech (Vuong 2016; Vuong and Napier 2015; Vuong et al. 2018a, 2021) as the impacts of religion on behaviors has been thoroughly discussed in the Vietnamese research community in the recent period (Vuong et al. 2019, 2020a). This can be illustrated briefly by the case of crowdfunding.

There are very few studies related to crowdfunding research. In the SSHPA database, there are only two papers (Nguyen et al. 2021a; Phan et al. 2020b) that are tangentially related to crowdfunding. Looking beyond this database, there is one master thesis (Nguyen 2017b) and a paper from the Asian Institute of Research (Do 2019). All studies only focus on the performance of the founders or the banks; thus, the depth of this area of research has not been dealt with properly in these studies.

In contrast, crowdfunding has been studied extensively in Thailand and Singapore since 2016. For example, Wonglimpiyarat (2017) provided evidence that crowdfunding is an effective way for the Thailand government to support startups in bridging the funding gap in the early stage. Adopting a similar position, Hu (2015) indicated that the Monetary Authority of Singapore approved the promotion of crowdfunding as a means for startups to approach funding. According to De Buysere et al. (2012), regulation, education, and research are the three main pillars to build a solid foundation for the development of crowdfunding. Recently, in February 2020, in Decision No. 283/QĐ-TTg, The Prime Minister of Vietnam decided to create the regulatory sandbox for some types of Fintech such as eKYC, digital wallet, and P2P (peer-to-peer) lending. Taken together, these results suggest the continual growth of research articles in fintech and regulations involving fintech.

### 4.3. Green Bank

Another topic that is hardly covered is green finance. Many nations have started to focus on the model of a green bank. According to Tran and Tran (2015), in definitive terms, a green bank is a banking model prioritizing environmental protection and sustainable development. Through its operations, the banking sector will serve as a link between

economic growth and environmental protection. With the initial objective to protect the environment by reducing the paper used in paperwork or applying new sources of clean energy, the banking sector in countries such as the United States, the United Kingdom, or India has expanded the green bank model to other activities of the economics, such as funding green projects, providing green products, encouraging green communications, etc. (Tran and Tran 2015). However, except for the success of the United States, the green bank model in other countries is yet able to generate significant impacts on the environment, or its impact is still under expectations, such as in the case of China (Bai et al. 2013). In Vietnam, the application of a green bank is a relatively new topic, and the number of research papers regarding this topic is very low. Applying green bank in a developing country such as Vietnam will yield various benefits, including avoiding paperwork, creating awareness to business people about the environment, or establishing loans at a lesser rate; thus, it is high time more aspects of green finance and green bank were needed to be elaborated and examined (Tran and Tran 2015). Researchers can delve into the expected cost and requirements for green bank application, and the obstacles for companies in taking advantage of the green finance movement, as well as policy suggestions for the government to better facilitate firms in green business (Vuong 2018, 2021).

## 5. Conclusions

In this study, we have systematically explored the Vietnamese financial landscape via a dataset of 314 international publications from 2008 to 2020. Among the first systematic reviews of Vietnam's financial research landscape, this study has shown empirical results from this key emerging market that underlined the importance of post-crisis regulation, which was evident in the control of non-performing loans and bank restructuring post-2008.

Moreover, over the past two decades, researchers have identified several immature aspects of the Vietnamese market, including human capital (Anwar and Nguyen 2010; Dinh et al. 2019), education level (Le et al. 2019a; Morgan and Long 2020; Van et al. 2019; Vo et al. 2019), and integration of technology (Dang et al. 2019; Huynh et al. 2018; Pham et al. 2018a). Improvement in these areas can not only increases Vietnamese market's attractiveness but also increase its capacity to absorb investment for further economic growth. In addition, our review also identifies the need for further studies on how to engender a shift of FDI flows from real estate to manufacturing sectors, and how to increase eco-friendly FDI to create a more sustainable financial market and distribute the economic gains of increased FDI equitably (Vuong 2021).

More importantly, there are three areas of research we expect to grow in prominence in the future. Firstly, because of the exceptional control of the COVID-19 outbreak, Vietnam's post-COVID-19 monetary regulations, and foreign trades, changes, and impacts of FDI on the domestic market will be a research area where a high level of research output is expected. Second, research on fintech and its regulations is also expected to grow, since recently, The Prime Minister of Vietnam decided to create the regulatory sandbox (Decision No. 283/QĐ-TTg) for certain types of fintech such as eKYC, digital wallet, and P2P (peer-to-peer) lending. Finally, for sustainable development, environmental protection, eco-friendly financial models should be promoted, thus launching a new wave of studies on green finance and green bank.

**Author Contributions:** Conceptualization, M.-T.H. (Manh-Tung Ho) and Q.-H.V.; methodology, M.-T.H. (Manh-Tung Ho); software, M.-T.H. (Manh-Toan Ho) and M.-H.N.; validation, M.-T.H. (Manh-Toan Ho) and Q.-H.V.; formal analysis, N.-T.B.L.; investigation, N.-T.B.L., Q.-H.N., H.-L.D.T.; resources, Q.-H.V.; data curation, M.-T.H. (Manh-Tung Ho), N.-T.B.L., Q.-H.N., and H.-L.D.T.; writing—original draft preparation, M.-T.H. (Manh-Tung Ho), N.-T.B.L., Q.-H.N., and H.-L.D.T.; writing—review and editing, M.-T.H. (Manh-Tung Ho), N.-T.B.L., Q.-H.N., M.-H.P., M.-H.L., and H.-L.D.T.; visualization, Q.-H.N.; supervision, M.-T.H.; project administration, Q.-H.V. All authors have read and agreed to the published version of the manuscript.

**Funding:** This research received no external funding.

**Institutional Review Board Statement:** Not applicable.

**Informed Consent Statement:** Not applicable.

**Data Availability Statement:** The dataset used in this research can be made available upon request.

**Conflicts of Interest:** The authors declare no conflict of interest.

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
