# Peer review of "A Systematic and Critical Review on the Research Landscape of Finance in Vietnam from 2008 to 2020"

_jrfm, doi:10.3390/jrfm14050219_

Round 1
Reviewer 1 Report
The manuscript is very interesting and is a review of an important topic for the discipline of finance, based on the Vietnam example. It is well-prepared and good cited. The article concerns the research landscape of finance research in Vietnam during the period 2008 to 2020.
An important advantage of the paper is also inclusion the analysis of the research undertake in Vietnam during the period of global coronavirus pandemic that makes the manuscript actual.
However, there are few main aspects that have to be improve if the Authors want to publish the paper in high-quality Journal:
1) The main objective states that the Authors also would like to predict the defining trends for future research direction, while there is a lack of the information about such prediction. They mentioned about the green finance, however these are they thoughts based on the world financial journals and articles, rather than a prediction. Moreover, green finance can not be the only one direction indicated. It has to be supplemented.
2) Furthermore, the Authors present the specific topics undertake in the analyzed papers in detailed, but they should also try to analyze or draw some conclusions from it, not only just describe them one by one. Even the article which is a review, has to have the in-depth analysis, especially if it will be published in such Journal. The simple description in not enough to be publish in JRFM.
3) Lack of conclusions. This is the biggest objection of the paper. Of course "discussion" provides some Authors’ considerations, however it is insufficient and the main conclusions have to be emphasized.
These free above imperfections have to be improved and supplemented before re-review.
Reviewer 2 Report
The paper touches on an interesting topic of the landscape of finance research in Vietnam during the period 2008 to 2020. The scientific approach applied in the paper is reasonable and accepted. Overall the paper construction is logical and it is mostly clearly written. The abstract is ok and informative enough. However, the paper has several drawbacks that need addressing.
- The instruction has to be significantly shortened and made more to the point. Now it includes selection of facts from Vietnam’s economic (recent) history. Please cut the redundant fragments from the introduction along with extensive references to data in the text. Leave information about the financial data trends in the analysed period. Also figure 1, table 1 should be dropped, while table 2 and figure 2 can be moved to appendix. Then merge what is left with 1.3.
Alternatively parts 1.1.-1.2 can be removed entirely without significant loss to the quality of the paper.
- Please add information about the structure of the paper at the end of section 1.3.
- Please add references to studies that have applied a similar methodology – reviews of trends in research papers. This is now missing. This can be done either in introduction or in methodology. Please do it briefly.
- Please justify (e.g. on the basis of other hitherto similar studies) in the paper the selection of keywords you use. Why only 6 keywords were chosen? Why other similar words, like e.g. “bank”, were omitted?
- Overall, although I appreciate Authors’ efforts in reviewing the cited papers, my overall impression is that the descriptions of trends in papers (in section 3) are too long and should be shortened where possible, to make them more to the point and don’t obscure the main points.
- The paper needs more visualisation of results. Please strongly consider doing in Excel stacked column charts (or charts of a similar type) using e.g. the data on number of papers in a given category, in a given year or calculate shares of papers of the given type etc.
- Please rename section 4 as “Discussion and conclusion” to better reflect its current content.
Please consider those comments as friendly suggestions to improve the quality of the paper. When submitting the revised version of the paper I kindly ask the Authors to consider providing a response letter explaining in detail how they reacted to comments. This will improve the timing of second round of review, if it is planned.
Reviewer 3 Report
The paper focuses on the research landscape of finance in Vietnam between 2008 and 2020. Basically, it is an interesting topic.
However, there are some structural and content problems in the text. The introduction isn't appropriate as it is in its current form a kind of mixture of materials and literature review, but in fact not an introduction. In the introduction, the authors should highlight the goal, importance, and context of the research, and the research questions should be described as well. So now the real introduction is completely missing. So the literature review. The paper doesn't contain a literature review in which the essential sources would be processed in a critical, analytical and comprehensive way.
The methodology the authors used is very poor, it is just going about a simple analysis of the Vietnamese national database for social sciences sources.
At the end of the paper there is a discussion chapter, but not the conclusions. Discussion should be earlier in the structure.
Based on the title we expect to read a systematic literature review but this paper in this form doesn't fulfill the criteria. It should be totally restructured and rewritten. In this form, I can't accept it for publication. Before resubmission, I highly recommend a professional English proofreading.
Round 2
Reviewer 1 Report
Dear Authors,
Well-done. All the presented in the review suggestions and comments were taken into considerations and included in the manuscript. I accept them. However, the only last one thing is that You did not present the paper’s contribution to the state of knowledge or to the general practice. It is also one of the key element of the high-quality article. You have to supplement it before publication.
Author Response
Dear Reviewer 1,
Thank you so much for your kind comments. We have added a statement for our contribution to the literature. Our paper is the only systematic review of Vietnam's finance research landscape. We have noted this contribution at the end of the Introduction part.
"Since there has been no comprehensive review of financial studies in Vietnam until now, this paper aims to contribute a systematic and critical review of key financial issues in one of the most dynamic emerging markets. Hence, providing not only research directions but also actionable insights for researchers and policy-makers, who seek to understand and strengthen financial systems in post-crisis financial markets."
We have also taken this opportunity to further tighten our texts. We have highlighted all changes on old texts in yellow, and newly added texts in green for your reference.
Once again, thank you so much for your kind comments and support. We hope our revisions have adequately addressed your concerns.
Best regards,
Reviewer 2 Report
I’ve thoroughly reviewed the revised version of the paper. All of my previous comments have been incorporated. The revised version has been much improved, and I thank the Authors for the constructive responses. The discussion on results was adequately updated, as requested. I recommend to accept the paper in the revised version as it is.
Author Response
Thank you so much for your comments and encouragement. We have also taken this opportunity to further tighten our texts. We have highlighted all changes on old texts in yellow, and newly added texts in green for your reference.
Best regards,
Reviewer 3 Report
The authors accepted my recommendations for improvements and did it well. Now the text is more clear, sampoling is better, scientifically it is more sound. I can accept it in this form for publication.
Author Response
Thank you so much for your kind comments and support. We have also taken this opportunity to further tighten our texts. We have highlighted all changes on old texts in yellow, and newly added texts in green for your reference.
We hope our revisions have adequately addressed your concerns.
Best regards,